# Enhanced Antifungal Activities of Eugenol-Entrapped Casein Nanoparticles against Anthracnose in Postharvest Fruits

**DOI:** 10.3390/nano9121777

**Published:** 2019-12-13

**Authors:** Yang Xue, Shitong Zhou, Chenyue Fan, Qizhen Du, Peng Jin

**Affiliations:** The Key Laboratory for Quality Improvement of Agricultural Products of Zhejiang Province, The College of Agricultural and Food Sciences, Zhejiang A & F University, Hangzhou 311300, China; gxgxy0205@163.com (Y.X.); zst18846494139@163.com (S.Z.); fffffcy1021@163.com (C.F.)

**Keywords:** essential oil, encapsulation, bovine casein, antifungal activity, fruit preservation

## Abstract

This study aims to improve the antifungal effects of eugenol through low-energy self-assembly fabrication and optimization of eugenol-casein nanoparticles (EC-NPs). Optimized EC-NPs (eugenol/casein ratio of 1:5) were obtained with a mean size of 307.4 ± 2.5 nm and entrapment efficiency of 86.3% ± 0.2%, and showed high stability under incubated at 20 and 37 °C for 48 h. EC-NPs exhibited satisfactory sustained-release effect at 20 °C or 37 °C, with remaining eugenols amounts of 79.51% and 53.41% after 72 h incubation, respectively, which were significantly higher than that of native eugenol (only 26.40% and 19.82% after the first 12 h). EC-NPs exhibited a greater antifungal activity (>95.7%) against spore germination of fungus that was greater than that of native eugenol, showed 100% inhibition of the anthracnose incidence in postharvest pear after 7 d. EC-NPs is potential as an environmental-friendly preservatives in the food industry.

## 1. Introduction

Up to now the low-cost and efficient of synthetic chemicals (i.e., inorganic salts, organic reagents, fungicides and pesticides) are widely used in the decay-controlling against postharvest diseases since the decay caused by spoilage microorganism invasion brings huge economic loss [1,2]. Considering to the food safety and human health, natural compounds that have low health or environmental impact are receiving wide attention [3,4]. Plant-derived bioactive substances have been applied as natural preservatives (e.g., essential oils, Base Natural, a commercial preservative) in the food industry due to their antifungal properties [5,6,7]. Eugenol (1,2-methoxy-4-(2-propenyl)-phenol), a principal component of the herbal essential oil from basil, has been classified as GRAS (generally recognized as safe) food additives, because reports indicated it has potential for the food field due to their antimicrobial properties against a wide range of microorganisms [8,9].

However, eugenol possesses high volatility and low water-solubility, which severely handicap its utilization due to the short timeliness for antibiosis [10]. In recent decades, the embedding and encapsulation has become a novel strategy to markedly offer numerous benefits for active compounds, for example, protection against oxidation, enhanced stability, retention of volatile ingredients and permeability [11]. Recently, encapsulation of eugenol showed enhanced bioactivity and stability [12,13]. However, the high-energy techniques or cumbersome processes for nano-formulation are difficult to scale-up for practical application. In addition, the preparation of some nanocarriers requires the addition of some chemical reagents, such as, dichloromethane [14], surfactant [15], which increases the risk of potential toxicity and safety that limit their applications in food industry [16]. Therefore, eco-safe technologies for the nano-formulation of eugenol are needed to develop, so that they can be used in the food industry [17,18].

Caseins, the major component of milk proteins in bovine milk, is a low-cost and commercially available food-grade additive in food and beverage [19]. In the native state of caseins, they are very stable for high temperature and pressure (for example, treated by 100 °C and 100 MPa without losing their essential integrity). The natural caseins can self-assemble to the micellar structure in solution, with diameters in the approximate range 10–400 nm [20]. The amphiphilic nature of the caseins causes them to act more as block copolymers micelles of alternating charge and hydrophobicity, which are suitable for encapsulating the compounds of poor water solubility in the hydrophobic core of the micelle [21]. Therefore, caseins have been widely used as efficient nanocarriers for hydrophobic drug for controlled release [22]. Accordingly, the present study aims to prepare eugenol-entrapped casein nanoparticles through a low-energy and simple self-emulsifying technique, which will provide a promising alternate for nano-formulation of eugenol against postharvest decay of fruits and vegetables.

## 2. Materials and Methods

### 2.1. Materials

Bovine casein (>98%) and eugenol (99%) were obtained from Sigma–Aldrich, Chemical Co. (St. Louis, MO, USA). Ethanol was purchased from China National Pharmaceutical Industry Co. Ltd. (Beijing, China). All solutions were prepared by using deionized water (Millipore, Bedford, MA, USA). All other reagents were of analytical grade.

### 2.2. Preparation of Nanoparticles

Bovine casein used for the preparation of nanoparticles was fully dissolved in deionized water to form a final concentration of 20 mg/mL. Eugenol was dissolved into ethanol with a final concentration of 200 mg/mL. The casein solution (10 mL) in 50 mL glass beaker was stirred for 30 min at 500 rpm, and then eugenol solution was stepwisely added into the casein solution with a volume of 5 μL. After the addition of eugenol solution, the mixture solution was continued to stir for 30 min to yield eugenol-casein nanoparticle (EC-NP) dispersion. The denatured protein in the dispersion was removed through centrifugation at 12,000 rpm for 10 min. The EC-NPs samples were stored in a freezer (4 °C) for further use.

### 2.3. Characterization of EC-NPs

The size distribution of the fresh nanoparticles was determined as mentioned in our previous study [23]. In brief, 500 μL of the nanoparticles suspension were diluted into 5 mL of pre-filtered deionized water. The analysis was performed in dynamic light scattering using a Zetasizer ZS 90 instrument (Malvern Instruments, Malvern, UK) at 25 °C temperature, employing a nominal 5 mW He−Ne laser operating at a 633 nm wavelength and 173° scattering angle. The EE was obtained by determining the free eugenol in EC-NPs solution, which was separated by using an Amicon Ultra-7K centrifugal filter device (7000 MWCO, Millipore Corp., Billerica, MA, USA.). The quantitative analysis of eugenol was performed by a high-performance liquid chromatographic (HPLC) Shimadzu LC 20A system (Shimadzu, Kyoto, Japan), consisted of two LC-10A pumps, an SIL-10Avp autosampler, an SPD-M10Avp UV detector and a Symmetry C18 (5 μm, 4.6 mm × 250 mm) column. The mobile phase was composed of methanol/water (65:35) at a constant flow rate of 1 mL/min at 30 °C, and monitored at 282 nm [24]. All data were expressed as the mean value of three independent batches of the samples.

The entrapment efficiency (EE, in percent) of eugenol was calculated as the percentage of entrapped eugenol to total eugenol (as the following equation).

$$\mathrm{EE}\%=\frac{\mathrm{Total}\;\mathrm{amount}\;\mathrm{of}\;\mathrm{eugenol}-\mathrm{free}\;\mathrm{eugenol}\;\mathrm{amount}\;}{\mathrm{Total}\;\mathrm{amount}\;\mathrm{of}\;\mathrm{eugenol}}\;\times 100\%$$

The morphological characterization of eugenol nanoparticles was performed by transmission electron microscopy (TEM), (Hitachi, H-9500E, Tokyo, Japan) [25]. Briefly, a dispersion of NPs diluted with pure water was adsorbed onto a carbon-coated formvar film that was attached to a metal specimen grid. Excess sample was removed through blotting and the grid was covered with a small drop of staining solution (2% *w/v* phosphotungstic acid). The staining solution was left on the grid for a few min and then the excess solution was drained. The sample was allowed to air dry thoroughly and was then examined using a transmission electron microscope.

### 2.4. Stability Assessments of EC-NPs

To investigate the effects of temperatures on stability of the EC-NPs dispersions, 5 mL samples solution were placed in glass vials and stored at 4 °C, 20 °C and 37 °C, respectively. The entrapment efficiency (EE) of the dispersions was measured at predetermined time intervals (0, 2, 4, 8, 12, 24, 48 and 72 h) for the stability evaluation of the nanoparticles. After a treatment of the EC-NPs dispersion containing 4 mg/mL eugenol and 20 mg/mL caseins, the released eugenol and EE of EC-NPs were measured according to the above method. A mixture solution containing 4 mg/mL eugenol and 20 mg/mL caseins was used for a control. All samples were measured in triplicate.

### 2.5. Cell Culture

Luria−Bertani (LB) medium (g/L; 10 g tryptone, 5 g yeast extract and 10 g NaCl) added 1% glucose was used for the isolation and culture of spoilage microorganisms. Spoilage bacteria *Botrytis cinerea* [26] was isolated from decayed pear fruits. Mycelium was precultured on LB agar media at 30 °C for 5 d as the seed. The spore suspension was prepared as our previous description [23], by washing the 20-day seed cultures with sterile water containing 0.01% (v/v) Tween-80 and then diluting them to 1 × 10^5^ spores/mL with the aid of a hemocytometer.

### 2.6. Determination of Antifungal Activity

The antifungal sensitivity of free eugenol and EC-NPs nanoparticles was determined by modified disc diffusion method [27]. Briefly, various amounts of free eugenol (1.00, 4.03, 8.04, 10.06, 15.08, 20.11, 25.13, 30.16, 35.19, 40.21 and 50.24 μg/mL) and equivalent eugenol-entrapped casein nanoparticles solution were uniformly smeared onto the LB agar media, respectively. Of the prepared spore suspension 5 μL was inoculated on the sterile filter paper at the center of the petri dish, and then incubated at 30 °C. The same manual tests were performed for control without eugenol. Finally, the inhibition rate of mycelium growth was calculated as a percentage of the control groups without eugenol. Minimum inhibitory concentration (MIC) is defined as the lowest concentration of eugenol or EC-NPs, which completely inhibits visible growth on solid media. To further quantitatively investigate the sustained-release effect of EC-NPs on the spore germination, the spore suspension was inoculated into 20 mL LB medium, added with an initial concentration of 40 μg/mL eugenol in both nanoencapsulated and nonencapsulated groups, respectively, and then cultured at 30 °C with shaking (200 rpm). The optical density at 600 nm (OD600) was used to monitor cell growth. The control was set without eugenol in the culture, and the entire experiment contained three replicates.

### 2.7. Effects of EC-NPs on Anthracnose Disease on Pear Fruit

The epidermis of pear fruits (a pear orchard in Hebei, China) was scrub with sterile saline, and then inoculated with 5 μL of *Botrytis cinerea* spore suspension (1 × 10^5^ spores/mL) in the middle of each fruit with a sterile injection needle. An equivalent amount of eugenol (4 mg) of free eugenol (1 mL) and EC-NPs solution (1 mL) was sprayed onto the pear fruit surface. Each group contained three replicates of 25 pear fruits, and the control group was correspondingly treated with sterile distilled water (1 mL). Subsequently, all of the treated samples were stored at 25 °C in a sterile room. Disease incidence and lesion diameters were recorded. The incidence was defined as the appearance of decay spots and the color changes of flesh at the inoculation hole. All data were analyzed by using Duncan’s multiple range tests in the SPSS 13.0 software (IBM Corp., Armonk, NY, USA)

## 3. Results and Discussion

### 3.1. Characterization of Eugenol-Entrapped Casein Nanoparticles

Casein nanocarriers are widely used as delivery systems for hydrophobic drugs and bioactive compounds due to its excellent emulsification [28] and self-assembled ability [22]. In this work, eugenol-loaded casein nanoparticles (EC-NPs) were prepared through the pre-formation of spherical casein micelles followed by the stirring treatment of the mixture with eugenol, resulting in assemblies of casein/eugenol complexes at the nanometric scale. The effects of the eugenol concentration on the particle size, polydispersity (PDI), zeta potential, and entrapment efficiency were further studied as shown in Table 1. The increase of the eugenol amount from 1 to 6 mg/mL resulted in formation of larger nanoparticles from 249.3 to 333.8 nm. Similar results were observed in previous studies, in which the presence of antimicrobial agent yielded a more viscous dispersed phase, resulting in larger particles [13,14]. Meantime, PDI values of casein nanoparticles was found to in the range of 0.261–0.333, which indicates moderate size distribution [29]. Size and size distribution are important characteristic indexes because it is related to the release of active compound. Generally, spherical shape and better dispersity are beneficial to the release of active compound [30]. Zeta potential is also an important parameter to reflect the physicochemical and biological stabilities of nanoparticles in dispersion [31], which helps the formulation to enhance the long-term stability [32]. As shown in Table 1, the addition of eugenol slightly increased negative charges on the particle surface from −14.47 to −21.91 mV, which indicated that the nanoparticles with eugenol concentration of 4 mg/mL exhibited ideal particle stability.

Entrapment efficiency (EE) is used to indicate the amount of compound entrapped into the polymeric matrix. The EE of the nanoparticles reached the maximum value of 91.1% when eugenol concentration was 2 mg/mL. However, with the eugenol rising from 2 to 6 mg/mL, the EE decreased from 91.1% to 67.1%. This result was in agreement with the nanoethosomes in our previous work [23]. The EC-NPs made of 2 mg/mL of eugenol exhibited a spherical shape (Figure 1a). Furthermore, to optimize the formulation of the nanoparticles, the antifungal effects of EC-NPs with consistent eugenol amount were evaluated in LB media against anthracnose strain *Botrytis cinerea*. As shown in Figure 1b, EC-NPs with 4 mg/mL eugenol showed the highest antifungal efficiency of 88.5% against anthracnose after 48 h of inoculation. There was a positive correlation between eugenol concentration and antifungal efficiency in this study. This result also showed that this increase in particle size did not compromise the antifungal action of the eugenol component-grafted EC-NPs.

### 3.2. Release Assessment of Eugenol Nanoparticles

In order to study the stability effects of encapsulated eugenol compared with that of native eugenol, the eugenol residual amount of EC-NPs and free eugenol in solution was investigated for incubation at different temperatures. As shown in Figure 2, EC-NPs was stable for 72 h when incubated at 4 °C or 20 °C, with still 88.75% and 79.51% of remaining eugenol in the EC-NPs, respectively. Even after 72 h incubation at 37 °C, the EC-NPs showed a satisfactory slow release trend, possessing 53.41% of remaining eugenol. However, the unencapsulated eugenol solution gradually lose eugenol at 4 °C, only 21.09% of remaining eugenol after 72 h. As to 20 and 37 °C, the native eugenol possessed remaining eugenol amounts of only 26.4% and 19.82% after 12 h, respectively, and the eugenol lose about 80% of the initial amount, and especially the eugenol disappeared almost after 48 h. This result further clearly confirmed that native eugenol usually occurs with the burst effect dissipation and resulted in the short-term existence, which greatly compromised its long-lived antifungal effect [23]. The higher amounts of eugenol remaining in solution containing EC-NPs lead to the better lasting antifungal effect, which implied that the encapsulation of eugenol into casein nanoparticles could improve the stability and produce maintain-released effect of eugenol. Furthermore, the particle size, PDI and zeta potential of EC-NPs were further investigated for incubation at different temperatures (Table 2). These results showed that EC-NPs exhibited remarkable physical and chemical stability at 20 °C for 48 h though the size and PDI values present slight increase.

### 3.3. Antifungal Assessment of Eugenol Nanoparticles

The antifungal activity of eugenol and its potency were quantitatively assessed by determining the MIC. Table 1 show the MIC of free eugenol and EC-NPs nanoparticles against anthracnose in vitro. The difference in the mycelium diameter of the inhibition zone indicates the sensitivity of fungal strains to various concentrations of free eugenol or EC-NPs. The MIC values of EC-NPs against anthracnose were 40.21 μg/mL, which were slightly lower than those of free eugenol (50.24 μg/mL), which indicate EC-NPs possessed higher antifungal activity than free eugenol (Table 3). These results further show the property of eugenol and EC-NPs, i.e., they are potential to inhibit the growth of anthracnose in common fruit. Compared with the free form of eugenol, the antifungal ability of EC-NPs was compromised at the initial period of antifungal reactions [33], due to the slow release profile of the nanoencapsulated eugenol. This result was consistent with our previous work on the antifungal effect of eugenol nanoethosomes [23].

Evaluation of antifungal ability of EC-NPs was also performed by observing its inhibition effect on the mycelial diameter of anthracnose on LB agarose plate. As shown in Figure 3a, after incubation for 48 h, the mycelial growth was observed with significant differences between the EC-NPs group and other groups. Compared with the control, native eugenol and casein-eugenol mixture gave 50% inhibition of the mycelial growth. In contrast, the EC-NPs showed a completely inhibition on the fungal growth. Furthermore, the antifungal effects of EC-NPs were evaluated by calculating the inhibition rate on spore germination of fruit *Botrytis cinerea* pathogen fungus. As shown in Figure 3b, after 24 h of incubation, the spore germination rate of native eugenol group and casein-eugenol mixture group already reached to 51.8% and 61.1%, respectively, while the spore germination rate of EC-NPs were only 4.3% after 48 h and 38.7% after 72 h. Obviously, the fungal inhibition effect of EC-NPs was stronger than the native eugenol or casein-eugenol mixture, indicating that the nano-formulation of eugenol could effectively improve the fungal inhibition effectiveness, which should be contributed by the maintain-release of eugenol from EC-NPs, which prolonged the action time of eugenol against fungi. In the previous work we have determined that native eugenol is usually volatile with a burst-effect release pattern, resulted in rapid dissipation during the initial period [23], while EC-NPs gave a maintain-release during a long time. In result, EC-NPs possess the potential for the antifungal application in post-harvest fruits.

### 3.4. EC-NPs as Preservative against Fruit Corruption

EC-NPs was further determined the effectiveness on suppressing the anthracnose of fruits caused by the pathogenic invasion. Compared to the native eugenol, the rot of pear fruit caused by *Botrytis cinerea* strain was employed to test the effectiveness. After spore suspension (10 μL, 1 × 10^5^ CFU/mL) was injected into epidermis of the pear fruit, the rot of pear fruit was observed by sprayed native eugenol or EC-NPs. As shown in Figure 4a, after being inoculated for 7 d, the obvious rots (black spot) were observed on the epidermis at the control group (treatment with sterilized water instead of preservatives), and minor wound decay occurred in native eugenol-treated group. In contrast, almost no any black spot and wound decay was observed in EC-NPs-treated group. After cutting open the inoculation site, we found that both the control group and native eugenol-treated group decayed in deep pulp while EC-NPs-treated group did not decay in deep pulp (Figure 4b). Obviously, EC-NPs completely suppressed the pathogen. As we known, nanoparticles possess excellent permeability for cortex [23,34]. Thus EC-NPs might get through the deeper pulp to deliver eugenol for antifungal. The disease incidences of control and native eugenol-treated groups reached 100% and 89% for 8 d incubation after inoculation, while that of EC-NPs group was only 23% (Figure 4c). In combination of the inhibition of the fruit rot and the disease incidences, we can conclude that EC-NPs significantly potentiate the antifungal efficacy of eugenol in inhibiting fruit anthracnose.

## 4. Conclusions

Eugenol was encapsulated in casein micelles by simple process without any other additives. A mass ratio of 5:1 of caseins/eugenol yielded the highest encapsulation efficiency and stability for eugenol-casein nanoparticles (EC-NPs). EC-NPs significantly improve the antifungal efficacy against anthracnose. These results indicate that EC-NPs nanoparticles could be used as an economical and simple-manufactured preservative for postharvest fruits against microbial spoilage.

## Figures and Tables

**Figure 1 nanomaterials-09-01777-f001:**
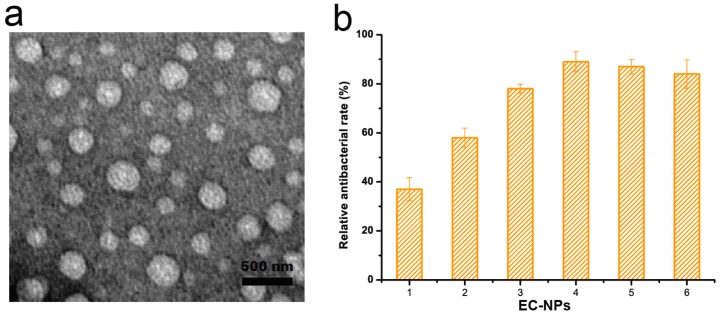
Characterization of eugenol nanoparticles. (**a**) Transmission electron microscopic image of eugenol-casein nanoparticles (EC-NPs). (**b**) Effects of eugenol concentration (mg/mL) on antifungal effect of eugenol-entrapped casein nanoparticles. The mean ± SD for three replicates are illustrated.

**Figure 2 nanomaterials-09-01777-f002:**
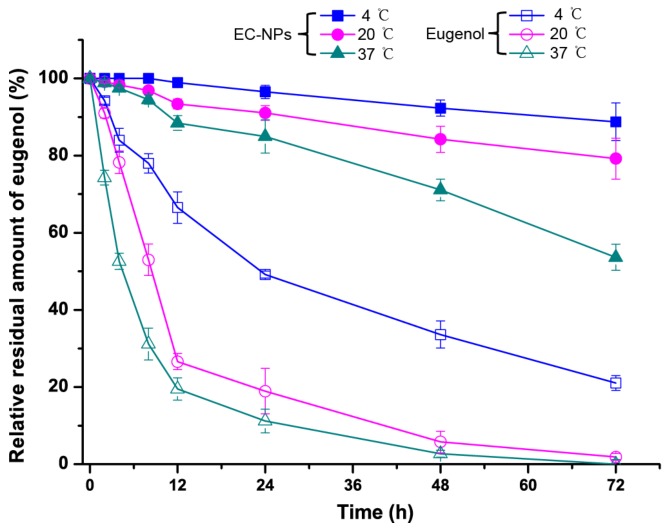
Effects of temperature on the eugenol stability in solution and in EC-NPs dispersion.

**Figure 3 nanomaterials-09-01777-f003:**
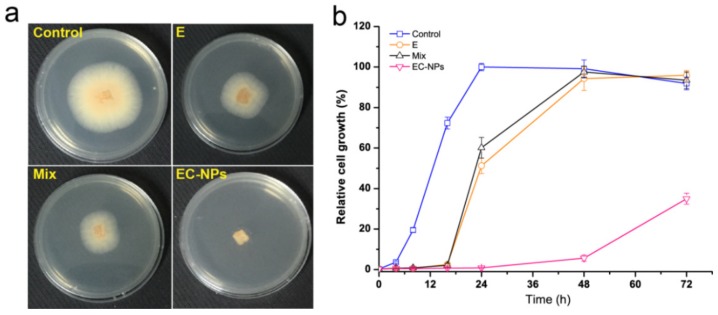
Evaluation of antifungal effects of eugenol nanoparticles. (**a**) Effects of native eugenol (E), casein-eugenol mixture (Mix) and eugenol-entrapped casein nanoparticles (EC-NPs) on the mycelial growth and (**b**) dynamic analyses of the spore germination in liquid media treated with the samples.

**Figure 4 nanomaterials-09-01777-f004:**
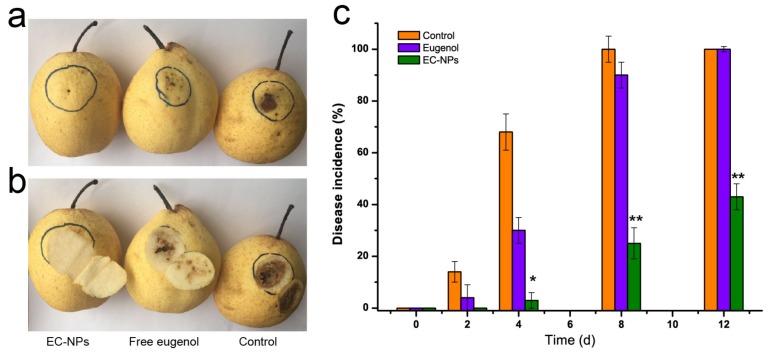
Effects of native eugenol and EC-NPs suppressing disease lesion of pear fruit inoculated with *Bo**trytis cinerea*. (**a**) Rot in the cuticular layer, (**b**) rot in pulp layer and (**c**) disease incidence of the pear fruit. Vertical bars represent the standard errors of the means of triplicate assays. The symbol * (*p* < 0.05) and ** (*p* < 0.01) indicate a significant difference between EC-NPs and native eugenol group.

**Table 1 nanomaterials-09-01777-t001:** Effects of the concentration of eugenol on the particle size, zeta potential, polydispersity indexes (PDI) and entrapment efficiency (EE) of eugenol loading in casein nanoparticles.

Eugenol (mg/mL)	Size (nm)	PDI	Zeta (mV)	EE (%)
1	249.3 ± 3.3	0.261 ± 0.003	−14.47 ± 0.31	90.4 ± 0.8
2	278.3 ± 5.1	0.270 ± 0.012	−16.61 ± 0.11	91.1 ± 0.3
3	289.6 ± 1.8	0.283 ± 0.024	−20.64 ± 0.23	87.1 ± 0.6
4	307.4 ± 2.5	0.284 ± 0.033	−21.18 ± 0.67	86.3 ± 0.2
5	333.8 ± 4.8	0.311 ± 0.009	−21.91 ± 0.37	70.1 ± 1.2
6	326.9 ± 6.3	0.333 ± 0.007	−20.20 ± 0.26	67.1 ± 0.9

**Table 2 nanomaterials-09-01777-t002:** Effects of the storage temperature on the particle size, zeta potential and polydispersity indexes (PDI) of EC-NPs nanoparticles after 48 h.

Temperature	Size (nm)	PDI	Zeta Potential (mV)
Control	307.4 ± 2.5	0.284 ± 0.033	−21.18 ± 0.67
4 °C	305.4 ± 1.9	0.289 ± 0.009	−21.01 ± 0.13
20 °C	310.3 ± 6.2	0.290 ± 0.008	−17.21 ± 0.32
37 °C	312.6 ± 4.2	0.303 ± 0.014	−17.32 ± 0.24

**Table 3 nanomaterials-09-01777-t003:** Effects of eugenol formulation with various concentrations on the mycelial growth inhibition.

C_Eugenol_ ^a^ (ug/mL)	Mycelial Growth Inhibition (%) ^b^		
	Native Eugenol	Casein-Eugenol Mixture	EC-NPs
1.00	0.00	0.00	0.00
4.03	1.88 ± 0.67	2.38 ± 1.02	1.47 ± 0.76
8.04	8.41 ± 1.35	7.32 ± 1.03	7.57 ± 0.87
10.06	9.41 ± 0.87	10.91 ± 2.17	9.24 ± 0.39
15.08	17.62 ± 2.23	15.62 ± 1.98	14.67 ± 2.21
20.11	24.97 ± 2.62	27.27 ± 2.01	31.48 ± 1.62
25.13	28.34 ± 3.12	30.34 ± 1.36	36.98 ± 0.28
30.16	38.70 ± 1.26	37.40 ± 3.72	44.40 ± 1.31
35.19	68.21 ± 1.67	69.93 ± 0.89	87.9 ± 2.34
40.21	87.43 ± 3.17	89.21 ± 2.34	100 ± 0.00
50.24	100 ± 0.00	100 ± 0.00	100 ± 0.00

^a^ The eugenol concentration is defined as all the eugenol amount in Luria–Bertani (LB) agar media. ^b^ The inhibition ratio is defined as a relative percentage of the control mycelial diameter after 72 h culture.

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
