# Peer review of "Enhanced Antifungal Activities of Eugenol-Entrapped Casein Nanoparticles against Anthracnose in Postharvest Fruits"

_nanomaterials, 2019, doi:10.3390/nano9121777_

Round 1

Reviewer 1 Report

The manuscript by Xue et al.  describes the encapsulation of eugenol in casein nanoparticles to generate a nanoparticle suspension that can act as fruit preservative. However, I have several concerns about it:

The writing in the introduction is not very clear, at least at the very beginning (lines 28-37). In addition, a deeper revision of the state of the art of the eugenol encapsulation will be interesting. Regarding Materials and Methods, more details should be provided. In section 2.3, there is no methodology for encapsulation efficiency, just an equation. In section 2.4, authors state that they are performing a stability and release study. In my opinion it should only say stability. If the capsules were perfect, no release would be observed. Authors should provide more information about the procedure. Were samples used dried or wet; in case of suspensions, what concentration... Results and Discussion. More in depth discussion and comparison with the work of other authors should be provided.
Table 1 and 2, What do you mean by Zate? Zeta Potential? It would be interesting to provide SEM images of the particles, to observe the surface, the presence or not of pores. In the stability test, where are the authors measuring the eugenol concentrations? It is clear to me, that if particles are not stable, they will be releasing eugenol to the aqueous medium in which they seem to be suspended. But, if the concentration in the aqueous medium also decreases, this means that eugenol is evaporated. Are the authors performing this assay in a closed system? In that case, it would be possible to measure the concentration of eugenol in the headpsace, in the liquid and in the particles and close the mass balance. In table 3, it is required to especify the units of the mycelial growth inhibition. In figure 3b, it is required to explain the legend. In section 3.4, it should be specified how the eugenol, the particles and the control were distributed over the fruit. Conclusions can be improved.  Authors should clarify that casein was used as encapsulating matrix, and this matrix stabilized the eugenol.  Authors claim that nanoparticles showed a greater encapsulation efficiency, compared to what? In addition, instead of slightly release effect, they should say stability

Author Response

Comments and Suggestions for Authors:The manuscript by Xue et al. describes the encapsulation of eugenol in casein nanoparticles to generate a nanoparticle suspension that can act as fruit preservative. However, I have several concerns about it:

Firstly, we sincerely appreciate the reviewer #1’ valuable suggestions and positive evaluation on our work. According to these comments and suggestions, we have carefully revised the manuscript which we hope meet with approval.

The writing in the introduction is not very clear, at least at the very beginning (lines 28-37). In addition, a deeper revision of the state of the art of the eugenol encapsulation will be interesting.

Accordingly, we have added the statements in the introduction of the revised manuscript, to make it clearer and interesting.

Regarding Materials and Methods, more details should be provided.

In section 2.3, there is no methodology for encapsulation efficiency, just an equation.

We have supplemented some detailed information in the revised Materials and Methods. Also we have given the methodology for encapsulation efficiency in 2.3 Section in the revised manuscript, and slightly modified the equation.

In section 2.4, authors state that they are performing a stability and release study. In my opinion it should only say stability. If the capsules were perfect, no release would be observed. Authors should provide more information about the procedure. Were samples used dried or wet; in case of suspensions, what concentration...

We have corrected this description with “stability” in revised manuscript. We have provided detailed information about the treating processes in the revised manuscript.

 Results and Discussion.

 More in depth discussion and comparison with the work of other authors should be provided. Table 1 and 2, What do you mean by Zate? Zeta Potential? It would be interesting to provide SEM images of the particles, to observe the surface, the presence or not of pores.

We have carefully revised the Results and Discussion section, and also added new references. We are very sorry for our spelling mistake on “Zeta Potential”. Accordingly, we have revised it with “Zeta” in the manuscript. This SEM image of the particles also could well reflect the spherical shape and particle size uniformity, and dimly observed the surface pores. We will further obtain higher resolution SEM images in the following work.

 In the stability test, where are the authors measuring the eugenol concentrations? It is clear to me, that if particles are not stable, they will be releasing eugenol to the aqueous medium in which they seem to be suspended. But, if the concentration in the aqueous medium also decreases, this means that eugenol is evaporated. Are the authors performing this assay in a closed system? In that case, it would be possible to measure the concentration of eugenol in the headspace, in the liquid and in the particles and close the mass balance.

All these solution for stability test were placed in glass vials (closed system) with negligible headspace. Therefore, the free eugenol in the solution can represent the accumulation of released eugenol from nanoparticles.

In table 3, it is required to especify the units of the mycelial growth inhibition.

We have added the % as the units of the mycelial growth inhibition in the revised manuscript.

In figure 3b, it is required to explain the legend.

We are very sorry for our mistake on the extra “(b)”. Accordingly, we have revised this legend.

In section 3.4, it should be specified how the eugenol, the particles and the control were distributed over the fruit.

We describe the details of these treatments in the method materials section 2.7, which specified how the eugenol, the particles and the control were distributed over the fruit.

Conclusions can be improved. Authors should clarify that casein was used as encapsulating matrix, and this matrix stabilized the eugenol. Authors claim that nanoparticles showed a greater encapsulation efficiency, compared to what? In addition, instead of slightly release effect, they should say stability

We have carefully rewritten the Conclusions, and corrected some incorrect statements in the revised manuscript.

Reviewer 2 Report

The authors did an interesting and useful research on the encapsulation of etheric oil on casein carriers and applied it to prevent the rotting of the fruit. This is a very efficient and low cost of usage of casein. 

I have no remarks or comments and I recommend acceptance in the present form. 

Author Response

Comments and Suggestions for Authors

The authors did an interesting and useful research on the encapsulation of etheric oil on casein carriers and applied it to prevent the rotting of the fruit. This is a very efficient and low cost of usage of casein.

I have no remarks or comments and I recommend acceptance in the present form.

We sincerely thank the reviewer #2’suggestions, positive evaluation and recommendation on our work. Those comments are all valuable and very helpful for revising and improving our paper, as well as the important guiding significance to our researches.

Reviewer 3 Report

This paper described the development of eugenol-entrapped casein nanoparticles with antibacterial activity against anthracnose in postharvest fruits. 

The manuscript is not well written and the English should be strongly improved.

Materials and Methods section:

Line 75: "Eugenol and ethanol were mixed...."Please, quantify the ratio (1:1 or 1:2 or 2:1?).

Paragraph 2.4: Investigate the stability and release of the EC-NPs dispersion at higher temperatures above 37°C (es. 45°C and 60°C) and also at lower temperatures such as -4°C and -10°C. These parameters can be important for the treatments that foods can undergo during their production and storage. Also can be important to investigate the stability and release of the EC-NPs dispersion at different pH, in particular at acid pH, as there are foods with pH <4.5. 

Paragraph 2.6: Please, insert a reference in the text for the antibacterial activity assay guidelines.

Results and Discussion section:

Paragraph 3.3:

Line 209:  Table1 does not show the MIC and MBC. Please, correct the mistake.

Line 210: "against various anthracnose microorganisms in vitro." In this paper,  the authors described the antibacterial activity only against Botrytis cinerea. Please, correct the mistake.

Author Response

Comments and Suggestions for Authors

This paper described the development of eugenol-entrapped casein nanoparticles with antibacterial activity against anthracnose in postharvest fruits.

The manuscript is not well written and the English should be strongly improved.

We sincerely appreciate reviewer #3 for the positive comments and good suggestions. We have carefully checked throughout the manuscript, and correct the mistakes in grammar, expression and spelling. Also we have asked a specialist with native English language to check our MS.

Materials and Methods section:

Line 75: "Eugenol and ethanol were mixed...."Please, quantify the ratio (1:1 or 1:2 or 2:1?).

We have rewritten the sentence to avoid misunderstanding.

Paragraph 2.4: Investigate the stability and release of the EC-NPs dispersion at higher temperatures above 37°C (es. 45°C and 60°C) and also at lower temperatures such as -4°C and -10°C. These parameters can be important for the treatments that foods can undergo during their production and storage. Also can be important to investigate the stability and release of the EC-NPs dispersion at different pH, in particular at acid pH, as there are foods with pH <4.5.

Our original goal is to apply casein-eugenol nanoparticles for the preservation of postharvest fruit and vegetables, which are mainly stored at normal or lower temperatures. Usually, the pH environment is not too low. Moreover, low pH may damage the caseins-nanoparticles since the isoelectric point of caseins is at pH 4.5-4.8. Therefore, we did not consider higher temperature and acid pH environments.

Paragraph 2.6: Please, insert a reference in the text for the antibacterial activity assay guidelines.

We have added a reference for this antibacterial activity assay guidelines.

Results and Discussion section:

Paragraph 3.3:

Line 209:  Table1 does not show the MIC and MBC. Please, correct the mistake.

We are very sorry for the incorrect description of MIC and MBC, and have rewritten this section on antimicrobial sensitivity assay in the revised manuscript. The MBC was deleted, and the MIC was defined as the lowest concentration of eugenol or EC-NPs which completely inhibits visible growth on solid media.

Line 210: "against various anthracnose microorganisms in vitro." In this paper,  the authors described the antibacterial activity only against Botrytis cinerea. Please, correct the mistake.

We have corrected this mistake.

Round 2

Reviewer 1 Report

The paper can be published in current form.

Reviewer 3 Report

The authors improved the manuscript following the reviewers' comments and revising the English language.